# Application of One-Step Reverse Transcription Droplet Digital PCR for Dengue Virus Detection and Quantification in Clinical Specimens

**DOI:** 10.3390/diagnostics11040639

**Published:** 2021-04-01

**Authors:** Dumrong Mairiang, Adisak Songjaeng, Prachya Hansuealueang, Yuwares Malila, Paphavee Lertsethtakarn, Sasikorn Silapong, Yongyuth Poolpanichupatam, Chonticha Klungthong, Kwanrutai Chin-Inmanu, Somchai Thiemmeca, Nattaya Tangthawornchaikul, Kanokwan Sriraksa, Wannee Limpitikul, Sirijitt Vasanawathana, Damon W. Ellison, Prida Malasit, Prapat Suriyaphol, Panisadee Avirutnan

**Affiliations:** 1Molecular Biology of Dengue and Flaviviruses Research Team, Medical Molecular Biotechnology Research Group, National Center for Genetic Engineering and Biotechnology, National Science and Technology Development Agency, Khlong Luang, Pathum Thani 12120, Thailand; dumrong.mai@biotec.or.th (D.M.); nattaya90@hotmail.com (N.T.); prida.mal@mahidol.ac.th (P.M.); 2Siriraj Center of Research Excellence in Dengue and Emerging Pathogens, Faculty of Medicine Siriraj Hospital, Mahidol University, Bangkok 10700, Thailand; adisak.son@mahidol.edu; 3Division of Dengue Hemorrhagic Fever Research, Faculty of Medicine Siriraj Hospital, Mahidol University, Bangkok 10700, Thailand; thiemmeca@gmail.com; 4Graduate Program in Medical Biochemistry and Molecular Biology, Department of Biochemistry, Faculty of Medicine Siriraj Hospital, Mahidol University, Bangkok 10700, Thailand; prachya.han@gmail.com; 5Food Biotechnology Research Team, National Center for Genetic Engineering and Biotechnology, National Science and Technology Development Agency, Khlong Luang, Pathum Thani 12120, Thailand; yuwares.mal@biotec.or.th; 6Department of Bacterial and Parasitic Diseases, Armed Forces Research Institute of Medical Sciences (AFRIMS), Bangkok 10400, Thailand; PaphaveeL.fsn@afrims.org (P.L.); sasikorns@afrims.org (S.S.); 7Department of Virology, Armed Forces Research Institute of Medical Sciences (AFRIMS), Bangkok 10400, Thailand; yongyuthp.fsn@afrims.org (Y.P.); ChontichaK@afrims.org (C.K.); damon.w.ellison.mil@mail.mil (D.W.E.); 8Division of Bioinformatics and Data Management for Research, Research Group and Research Network Division, Research Department, Faculty of Medicine Siriraj Hospital, Mahidol University, Bangkok 10700, Thailand; kwanrutai.chi@mahidol.edu; 9Graduate Program in Immunology, Department of Immunology, Faculty of Medicine Siriraj Hospital, Mahidol University, Bangkok 10700, Thailand; 10Pediatric Department, Khon Kaen Hospital, Ministry of Health, Khon Kaen 40000, Thailand; kanok.sriruksa@gmail.com (K.S.); sirijitt@gmail.com (S.V.); 11Pediatric Department, Songkhla Hospital, Ministry of Health, Songkhla 90100, Thailand; nunee074@hotmail.com

**Keywords:** dengue virus, droplet digital PCR, virus detection, virus quantification

## Abstract

Detection and quantification of viruses in laboratory and clinical samples are standard assays in dengue virus (DENV) studies. The quantitative reverse transcription polymerase chain reaction (qRT-PCR) is considered to be the standard for DENV detection and quantification due to its high sensitivity. However, qRT-PCR offers only quantification relative to a standard curve and consists of several “in-house” components resulting in interlaboratory variations. We developed and optimized a protocol for applying one-step RT-droplet digital PCR (RT-ddPCR) for DENV detection and quantification. The lower limit of detection (LLOD95) and the lower limit of quantification (LLOQ) for RT-ddPCR were estimated to be 1.851 log10-copies/reaction and 2.337 log10-copies/reaction, respectively. The sensitivity of RT-ddPCR was found to be superior to qRT-PCR (94.87% vs. 90.38%, *p* = 0.039) while no false positives were detected. Quantification of DENV in clinical samples was independently performed in three laboratories showing interlaboratory variations with biases <0.5 log10-copies/mL. The RT-ddPCR protocol presented here could help harmonize DENV quantification results and improve findings in the field such as identifying a DENV titer threshold correlating with disease severity.

## 1. Introduction

Dengue virus (DENV) is the causative agent of dengue fever (DF), dengue hemorrhagic fever (DHF), and dengue shock syndrome (DSS) [1]. There are approximately 360 million DENV infections resulting in 96 million symptomatic cases annually [2]. Currently, there are no anti-DENV therapeutics, and the commercially available vaccine has limited efficacy with an elevated risk of hospitalization in DENV seronegative vaccinees [3]. Adequate fluid management and supportive care are the only available means to treat DHF/DSS patients and are critical for a favorable outcome [1]. Thus, an accurate diagnostic tool could help healthcare personnel rapidly detect dengue in patients and promptly provide appropriate care.

Currently, rapid diagnostic tests (RDTs) based on detection of DENV non-structural protein 1 (NS1) or anti-DENV antibodies are widely used for screening dengue patients [4]. RDTs are inexpensive, and results are interpretable within a few minutes. However, the sensitivity and specificity of RDTs are limited [5,6]. RT-PCR is the standard method for confirming DENV infection because of its high sensitivity and specificity as well as a relatively short turnaround time compared to other methods such as virus isolation [1]. Thus, RT-PCR is usually used for validating novel DENV diagnostic tests including RDTs [7]. The assay also is used to quantify DENV RNA in laboratory and clinical samples to assess virus burden [8] and to determine the efficacy of vaccines [9] and anti-DENV drugs [8].

Although applications of RT-PCR and quantitative RT-PCR (qRT-PCR) using fluorogenic probes and in vitro transcribed RNA standards to detect and quantify DENV are common, there is no standardized protocol, making it difficult to compare results between laboratories [7,9]. Each laboratory often has “in-house” components in the protocol including primer sequences and PCR cycling profiles. One component which further complicates inter-laboratory comparisons is a standard curve. The standard curve is usually generated for each run, so errors may occur during the distribution of RNA standard and the preparation of serial dilutions [10]. In addition, standard RNA stocks must be continuously generated, checked, and maintained in the laboratory which requires expensive resources and well-trained staff [10].

In the past decade, digital PCR (dPCR) and droplet digital PCR (ddPCR) with the capability of quantifying nucleic acids without the use of a standard curve have been introduced. The techniques are based on discrete partitions of reactions in which either no or few molecules of nucleic acid templates are present [11,12]. Thus, the fraction of partitions with a signal at the end of the assay could be calculated as the number of template molecules. Further development of ddPCR to include reverse transcription enabled a modified protocol, called one-step reverse transcription droplet digital PCR (RT-ddPCR), to quantify RNA templates [13]. Thus, RT-ddPCR was applied for RNA quantification assays including enumerating genome copies of RNA viruses [14,15,16]. A recent study by Abachin et al. proposed and validated an RT-ddPCR protocol for quantitating DENV serotype 2 [10]. In this study, we optimized an RT-ddPCR protocol for the detection and quantification of DENV RNA of all four serotypes in clinical samples. In addition, we assessed inter-laboratory variations by performing DENV RNA quantification using the identical protocol in three laboratories.

## 2. Materials and Methods

### 2.1. RNA Standard Preparation

The 3′untranslated region (3′UTR) conserved region (base positions 10261-10620, Appendix A) of dengue serotype 4 (DENV4) was amplified and cloned into a plasmid (pGem) using the TA cloning system (Promega, Madison, WI, USA). The derived plasmid, pGem3UTRDv4-M14931, was transformed into E. coli, selected with ampicillin drug-resistance and confirmed by colony PCR assays. The plasmid containing the 3′UTR region was then propagated in LB broth (5 mL) containing ampicillin (100 µg/mL). The propagated plasmid was purified using the QIAprep plasmid mini prep kit (QIAGEN, Hilden, Germany). Then, 5 µg of purified plasmid was linearized using SacI restriction endonuclease (New England Biolabs, Ipswich, MA, USA) and cleaned up using the QIAquick gel extraction kit (QIAGEN, Hilden, Germany). Up to 0.5–1 µg of linearized plasmid was used as a template for in vitro RNA synthesis using the RIBOMAX^®^ T7 RNA production system (Promega, Madison, WI, USA) according to the manufacturer’s instructions. After in vitro transcription, the plasmid templates were removed by RNase free DNase I (Promega, Madison, WI, USA) treatment. The in vitro transcribed RNA was purified using the RNeasy Mini Kit (QIAGEN, Hilden, Germany). The quality of RNA was evaluated by Bioanalyzer^®^ (Agilent Technologies, Santa Clara, CA, USA). The concentration of RNA was obtained by Qubit^®^ 2.0 fluorometer with Qubit^®^ RNA BR Assay Kit (Thermo Fisher Scientific, Waltham, MA, USA). Finally, the concentrations of in vitro transcribed RNA standards were calculated as copy number/volume using EndMemo DNA/RNA copy number calculator (http://www.endmemo.com/bio/dnacopynum.php (accessed on 2 February 2021)) [17].

### 2.2. Clinical Specimens

Clinical specimens were EDTA-treated plasma samples retrieved from pediatric dengue clinical cohort sites in Khon Kaen Hospital and Songkhla Hospital collected between 2006–2013. The specimens were from 156 dengue patients (40, 40, 40, and 36 patients for DENV1, DENV2, DENV3, and DENV4, respectively) and 40 non-dengue patients. DENV infection was confirmed by “gold standard” techniques including IgM/IgG ratio from IgG/M-capture ELISA [18] combined with nested RT-PCR [19]. The serotype of DENV was identified by nested RT-PCR and serotype-specific NS1 ELISA [20]. Specimens with negative results from nested RT-PCR and IgG/M-capture ELISA were classified as other febrile illnesses (OFI).

DENV RNA was extracted from 140 µL of each plasma sample by using an automatic extraction ExiPrepTM Dx Viral RNA Kit (Bioneer, Daejeon, Korea) according to the manufacturer’s protocol resulting in a 50 µL elution volume. To prevent RNA degradation from multiple freeze/thaw cycles, the eluted RNA samples were aliquoted into small volumes (25 µL/tube) before storage at −70 °C until use.

### 2.3. Conventional qRT-PCR for Dengue Virus Detection and Quantification

Each qRT-PCR was carried out in 12.5 µL of one-step qRT-PCR Brilliant III Ultra-Fast probe master mix (Agilent Technologies, Santa Clara, CA, USA), containing 5 µL of the template RNA, following the manufacturer’s protocol. The sequences of forward and reverse primers and the probe specific to 3′-UTR for all DENV serotypes were 5′-GGTTAGAGGAGACCCCTCCC-3′, 5′-GGCGYTCTGTGCCTGGA-3′ and 5′-CAGGATCTCTGGTCTYTCCCAGCGT-3′ (Y represents a degenerate base, pyrimidine), respectively (Songjaeng, Thiemmeca and Avirutnan et al., submitted manuscript, Division of Dengue Hemorrhagic Fever Research, Faculty of Medicine Siriraj Hospital, Mahidol University). A quencher-labelled hydrolysis probe, 5′FAM/3′Tamra, was used (Integrated DNA Technologies, Coralville, IA, USA). The amplification reactions were performed in a LightCycler 480 II Thermocycler (Roche, Basel, Switzerland) detection system. The reactions included reverse transcription at 50 °C for 30 min, polymerase activation at 95 °C for 5 min, and PCR amplification for 45 cycles of 95 °C for 15 sec, 59 °C for 1 min (annealing-extension), and signal acquisition.

### 2.4. One-Step RT-ddPCR for Dengue Virus Detection and Quantification

The RT-ddPCR amplification reaction was performed according to the manufacturer’s protocol (one-step RT-ddPCR advanced kit for a probe, Bio-Rad, Hercules, CA, USA). The reaction consisted of 5 µL of the RNA template, 1X super mix master (Bio-Rad, Hercules, CA, USA), 40 U of reverse transcriptase enzyme, 300 nM of DTT, 0.9 µM of each forward and reverse primer, and 0.25 µM of hydrolysis probe. The sequences of forward and reverse primers and probe were identical to those used in conventional qRT-PCR described above (Songjaeng, Thiemmeca and Avirutnan et al., submitted manuscript, Division of Dengue Hemorrhagic Fever Research, Faculty of Medicine Siriraj Hospital, Mahidol University). The final volume of the reaction was adjusted to 20 µL with nuclease-free water. Next, the PCR reaction mix was converted to small droplets in an oil mixture by QX100TM Droplet Generator (Bio-Rad, Hercules, CA, USA). The droplet-in-oil mixture was then carefully loaded onto a 96-well PCR amplification plate. The PCR cycling was performed using a T100TM PCR thermal cycler (Bio-Rad, Hercules, CA, USA) by following the RT-ddPCR protocol. The PCR cycling steps were reverse transcription for 30 min at 60 °C, DNA polymerase activation for 5 min at 95 °C, and continuing to 40 amplification cycles of 94 °C for 30 s and 55 °C for 1 min/cycle followed by the inactivation of enzyme mix at 98 °C for 10 min. Sterile molecular grade water was used instead of the RNA template in the negative control reaction. Finally, the droplets containing PCR products in the 96-well plate were analyzed for signals using the Bio-Rad QX100 droplet reader. Absolute quantification of the PCR target was analyzed using QuantaSoftTM software version 1.7.4.0917 (Bio-Rad, Hercules, CA, USA).

### 2.5. Optimizing DENV RT-ddPCR Conditions

Gradient PCR was used to optimize the annealing temperature. In vitro transcribed RNA standard was diluted to 10^5^ copies/μL and used as a template. Two different quencher-labeling of hydrolysis probes, 5′FAM/3′Tamra and 5′FAM/3′BHQ quencher (Integrated DNA Technologies, Coralville, IA, USA), were also tested for efficiency. The PCR cycling steps were reverse transcription for 30 min at 60 °C; DNA polymerase activation for 5 min at 95 °C continuing to 40 amplification cycles of 30 sec at 94 °C; in each row, the annealing/extension temperature was adjusted to 64.0 °C, 63.2 °C, 61.8 °C, 59.7 °C, 57.2 °C, 55.1 °C or 53.8 °C for 1 min; followed by 10 min at 98 °C for enzyme heat deactivation. The negative control consisted of nuclease-free water instead of the RNA sample.

### 2.6. Lower Limit of Detection (LLOD95) and Lower Limit of Quantification (LLOQ)

The LLOD95 and the LLOQ of RT-ddPCR were assessed by using the in vitro transcribed RNA standard described above and diluted serially in sterile nuclease-free water. Seven expected concentrations of RNA standards ranging from 1 × 10^5^ (undiluted), 1 × 10^4^ (1:10), 1 × 10^3^ (1:100), 1 × 10^2^ (1:1000), 50 (1:2000), 10 (1:10,000), and 5 (1:20,000) copies/reactions were detected by RT-ddPCR following the protocol described above. The RT-ddPCR of each dilution was performed in triplicate, and the RT-ddPCR was independently repeated three times. Thus, a total of nine measurements were performed for each dilution.

### 2.7. Inter-Laboratory Variations

For the inter-laboratory comparison, 60 clinical specimens were randomly selected from the 196 specimens mentioned above. The selected 60 specimens consisted of 15 samples per DENV serotype. The DENV RNA samples were extracted and distributed to three external laboratories in Thailand including (1) Department of Virology, Armed Forces Research Institute of Medical Sciences (AFRIMS) for qRT-PCR (Hydrolysis) assay, (2) Department of Bacterial and Parasitic Diseases, AFRIMS for RT-ddPCR, and (3) Food Biotechnology Research Unit, National Center for Genetic Engineering and Biotechnology (BIOTEC, Pathum Thani) for RT-ddPCR. qRT-PCR (Hydrolysis) assay performed by AFRIMS was based on a protocol described elsewhere [21,22]. AFRIMS and BIOTEC laboratories repeated RT-ddPCR assays using the reagents and protocols optimized in this study.

### 2.8. Statistical Analysis

The number of specimens in this study was calculated for validating the sensitivity and specificity of one-step RT-ddPCR in DENV RNA detection. The sensitivity and specificity were expected to be at least 90% with margins of error of ±10%. Thus, at least 35 specimens/test group were required to validate the sensitivity and the specificity with a confidence level of 95% (α = 0.05) [23]. To evaluate the serotype-specific sensitivity, 36–40 specimens/serotype were retrieved as mentioned above. Additionally, 40 specimens from OFI patients were retrieved to evaluate the specificity. The LLOD was used to set a threshold for positive results for both qRT-PCR and RT-ddPCR. Exact McNemar’s test was used to compare sensitivity and specificity between qRT-PCR and RT-ddPCR [24].

In this study, LLOD95 and LLOQ were determined by quantitating serially diluted in vitro transcribed RNA standards. LLOD95 was defined as the lowest concentration at which RT-ddPCR had a probability for detecting DENV RNA in at least 95% of repeated measurements. LLOD95 was estimated by logistic regression analysis as previously described [25,26]. LLOQ was defined as the lowest concentration greater or equal to LLOD95 at which RT-ddPCR reported quantities from all repeated measurements with the coefficient of variation (%CV) equal or less than 20% [27]. In addition, the reported concentration of LLOQ must be within a ± 20% range of the anticipated concentration [27], which was calculated as an average of log10-transformed reported concentrations of the undiluted standard multiplied by corresponding dilution factors [26]. To estimate LLOQ, a precision profile was generated, and the lowest concentration corresponding to %CV = 20% was estimated with an exact CLSI EP17 model with “VFP” package of R program. In addition, a simple linear regression analysis was used to fit reported concentrations above LLOD95 to estimate the lowest concentration with a bias within ± 20% range of the corresponding anticipated concentration.

To analyze inter-laboratory variations and compare results between qRT-PCR and RT-ddPCR, linear regression, Pearson correlation, and Bland–Altman plots were used. Reported concentrations above ULOQ or below LLOQ were removed from analyses.

R programming language (version 3.6.1, The R Foundation for Statistical Computing, Vienna, Austria) was used for all statistical analyses and graph plotting.

## 3. Results

### 3.1. Optimizing RT-ddPCR Protocol for DENV RNA Quantification

Since no RT-ddPCR protocol for DENV was available at the beginning of this study, the RT-ddPCR protocol from the manufacturer was tested. However, the separation of signals between the positive and negative droplets was insufficient to set the quantification threshold (data not shown) so the protocol needed to be optimized. First, the concentration of primers was varied from 900 nM to 300 nM, but the separation of signals was not improved (data not shown). Second, a reaction with 250 nM of FAM/Tamra probe was compared to a reaction with 250 nM of FAM/BHQ probe recommended by the manufacturer. FAM/Tamra probe improved the signal separation compared to the original protocol as shown in Figure 1. Thus, FAM/Tamra-labeled probe was used in this study. Next, an optimal annealing temperature was determined by gradient PCR using a temperature range from 53.8 °C to 64.0 °C. The annealing temperature at 53.8 °C produced the widest signal separation (Figure 2). However, the annealing temperature at 53.8 °C was not recommended by the manufacturer due to potential primer-dimer formation. Thus, the annealing temperature was set at 55 °C for this study.

### 3.2. Lower Limit of Detection (LLOD) and Lower Limit of Quantification (LLOQ)

Although the LLOD of ddPCR was one template per 20 µL reaction according to the manufacturer, LLOD and LLOQ for DENV RNA were not established. Thus, in vitro transcribed RNA standards were analyzed to determine these limits. For quality control, repeated runs were planned for reactions with less than 10,000 acceptable droplets. Three independent runs with triplicate reactions per run yielded nine repeated measurements for each concentration of RNA standard (Appendix A). There was no reaction that yielded less than 10,000 acceptable droplets so repeated runs were not required.

To estimate LLOD based on a probability of detection at 95% (LLOD95), a logistic regression analysis was performed (Figure 3A). An estimated LLOD95 corresponded to a 1:663 dilution. Using an anticipated concentration of the undiluted sample (i.e., 4.671 log10-copies/reaction or 46,927.06 copies/reaction) for template count conversion, the LLOD95 was 1.851 log10-copies/reaction or 70.91 copies/reaction. To determine LLOQ, the lowest concentration with a coefficient of variation (%CV) = 20% and a bias within a ±20% range of the corresponding anticipated concentration was estimated. The precision profile (Figure 3B) showed that an estimated %CV was 20% at 1.090 log10-copies/reaction. However, this concentration was less than the LLOD95 so it was not an appropriate LLOQ. Next, linear regression was used to estimate the lowest concentration with a bias within a ±20% range of the corresponding anticipated concentration (Figure 3C). The estimated LLOQ was 2.337 (95%CI 2.286–2.389) log10-copies/reaction or 217.02 copies/reaction. An estimated %CV from the precision profile (Figure 3B) at the LLOQ was 5.9%. Based on the RNA extraction protocol, LLOD95 and LLOQ mentioned above as copies/reaction were equivalent to 3.705 and 4.190 log10-copies/mL of the plasma specimen, respectively. According to the manufacturer, the upper limit of quantification (ULOQ) was 100,000 copies/reaction which was equivalent to 6.854 log10-copies/mL of the plasma specimen.

### 3.3. Sensitivity and Specificity

The diagnostic performances of RT-ddPCR and qRT-PCR were determined with 196 clinical samples including 156 samples from dengue patients (40 DENV1, 40 DENV2, 40 DENV3, and 36 DENV4) and 40 samples from other febrile illness (OFI) patients (Table 1). The RT-ddPCR technique showed a sensitivity of 94.87% (95%CI 90.21%–97.38%) and specificity of 100.00% (95%CI 91.24%–100.00%) while qRT-PCR showed a sensitivity of 90.38% (95%CI 84.74%–94.09%) and specificity of 100.00% (95%CI 91.24%–100.00%). The sensitivity of RT-ddPCR was significantly higher than that of qRT-PCR (exact McNemar’s test *p* = 0.039). In addition, serotype-specific analyses showed that the sensitivities of RT-ddPCR were higher than those of qRT-PCR for DENV1 (97.50% versus 95.00%) and DENV3 (90.00% versus 75.00%, Appendix A). However, the difference was not deemed significant due to small sample sizes. The sensitivities for detecting DENV2 (92.50%) and DENV4 (100.00%) were identical for RT-ddPCR and qRT-PCR.

### 3.4. Determination of the Variation Between RT-ddPCR and qRT-PCR

Since the RT-ddPCR protocol was developed based on our qRT-PCR protocol (Songjaeng, Thiemmeca and Avirutnan et al., submitted manuscript, Division of Dengue Hemorrhagic Fever Research, Faculty of Medicine Siriraj Hospital, Mahidol University), several components were shared including the target sequences. Thus, the quantification results from the two protocols conducted in the same laboratory were expected to be similar. To compare quantification results, the experiments were done with 156 dengue specimens used in the sensitivity analyses. The quantification results from the two methods strongly correlated (Pearson’s *r* = 0.8538, *p* < 0.001, Figure 4A). The Bland–Altman plot showed that RT-ddPCR reported fewer RNA counts with a bias of −0.342 ± 0.446 log10-copies/mL, and the limit of agreement was −1.217 to 0.532 log10-copies/mL (Figure 4B). The linear regression analysis of the Bland–Altman plot showed a moderate negative correlation (Pearson’s *r* = −0.5555, *p* < 0.001) meaning that there was a significant trend of bias in which RT-ddPCR would report far fewer RNA counts than qRT-PCR as the RNA count increased. Serotype-specific analyses also showed strong correlations of quantification results (Appendix A). DENV1 had the lowest correlation (Pearson’s *r* = 0.7415, *p* < 0.001) while DENV4 had the highest correlation (Pearson’s *r* = 0.9227, *p* < 0.001). The biases ranged from −0.381 ± 0.309 log10-copies/mL for DENV2 to −0.308 ± 0.395 log10-copies/mL for DENV4 (Appendix A). The limits of agreement ranged from −0.986 to 0.224 log10-copies/mL for DENV2 to −1.413 to 0.714 log10-copies/mL for DENV1 (Appendix A). The moderate negative correlations in Bland–Altman plots were also observed in all four serotypes.

### 3.5. Determination of the Quantification Variations Between Laboratories

To determine interlaboratory variations of DENV RNA quantification by RT-ddPCR, the DENV RNA was measured in 60 clinical samples by three laboratories in Thailand including our laboratory (Division of Dengue Hemorrhagic Fever Research, Siriraj Hospital, SI), Armed Forces Research Institute of Medical Sciences (AFRIMS), and National Center for Genetic Engineering and Biotechnology (BIOTEC). Quantification results from those three laboratories were subsequently compared based on regression lines and Bland–Altman plots (Figure 5 and Figure 6). According to the regression analyses, the results between SI and BIOTEC yielded the highest correlation with Pearson’s *r* = 0.9308 (*p* < 0.001, Figure 5C). The Bland–Altman plots also showed the smallest bias (−0.028 ± 0.215 log10-copies/mL) and the narrowest limit of agreement (−0.451 to 0.394 log10-copies/mL, Figure 6C). The results from AFRIMS and BIOTEC showed the lowest correlation with Pearson’s *r* = 0.8758 (*p* < 0.001, Figure 5B) and the widest limit of agreement of −0.860 to 0.229 log10-copies/mL (Figure 6B). Interestingly, regression analyses of Bland–Altman plots showed no significant correlations between SI and AFRIMS (Pearson’s *r* = 0.0141, *p* = 0.922), BIOTEC and AFRIMS (Pearson’s *r* = 0.0224, *p* = 0.877), and SI and BIOTEC (Pearson’s *r* = −0.0987, *p* = 0.491). Thus, interlaboratory variations of RT-ddPCR appeared to be random and unrelated to the RNA counts in contrast to variations between qRT-PCR and RT-ddPCR. We also quantified the clinical samples mentioned above by qRT-PCR at our laboratory (SI) and AFRIMS (Appendix A). An increase in interlaboratory variations was observed as evidenced by the decrease in the correlation of quantification results (Pearson’s *r* = 0.6519, *p* < 0.001) and the widening of the limit of agreement range (−2.143 to 1.746 log10-copies/mL). Notably, SI and AFRIMS qRT-PCR protocols were fundamentally different including different target sequences which could influence the assay performance (Songjaeng, Thiemmeca and Avirutnan et al., submitted manuscript). Thus, the interlaboratory variations of qRT-PCR shown here could not be solely attributed to factors/errors introduced during the process of the assay and should not be directly compared to the variations of RT-ddPCR.

## 4. Discussion

Although qRT-PCR is a conventional method for quantifying DENV RNA, the results reported from different laboratories are difficult to compare due to interlaboratory variations. We demonstrated these variations by performing qRT-PCR with identical sets of clinical specimens in two different laboratories using two different protocols (Appendix A). This experiment imitated an actual situation in which each laboratory had developed its own in-house qRT-PCR protocols as adopting a protocol from another laboratory is complicated and often impractical. Since it was not possible to extract the effect of the assay errors from the effect of the difference in protocols on the interlaboratory variations, the RT-ddPCR protocol reported here was not primarily intended to reduce the assay errors compared to qRT-PCR. Instead, the protocol was designed to be reproducible in other laboratories with no or limited experience in quantifying virus RNA.

Although Abachin et al. reported a similar one-step RT-ddPCR technique, their protocol was validated only with DENV2 RNA from laboratory samples [10]. In our study, we have extended our protocol for the detection and quantification of all four DENV serotypes by targeting a conserved region at the 3′UTR end of the genome. The sequences of primers and probes were checked against all DENV genomes reported in GenBank to ensure that viruses collected in 2013 or earlier were detected. In addition, the primers and probes were tested against the extracted viral RNA from cultures of other flaviviruses including Japanese encephalitis virus, yellow fever virus, and zika virus using conventional RT-PCR and qRT-PCR yielding no false-positive results (Songjaeng, Thiemmeca and Avirutnan et al., submitted manuscript, Division of Dengue Hemorrhagic Fever Research, Faculty of Medicine Siriraj Hospital, Mahidol University). Initially, the standard protocol recommended by the ddPCR manufacturer was tested. However, we found that FAM/BHQ-labeled probes recommended by the manufacturer and used in another RNA virus study [26] yielded fluctuating signals from negative droplets in control reactions (Figure 1). We suspected that dithiothreitol (DTT), a reducing agent in the RT-ddPCR kit, accelerated the degradation of BHQ imitating positive signals. DTT was found to disturb real-time PCR by inducing quenching of the passive reference signal resulting in an overestimation of DNA concentrations [28]. Thus, FAM/Tamra-labeled probes were then selected based on a previous quantification study in the hepatitis E virus [29].

To evaluate the performance of RT-ddPCR, the limit of detection (LOD) and the limit of quantification (LOQ) must first be established. Ideally, synthetic DENV RNA standards with known absolute quantity would be used for determining LOD and LOQ. However, commercially available DENV RNA standards (ATCC, Manassas, VA, USA) do not contain the conserved 3′UTR region. Therefore, in vitro transcribed DENV RNA routinely used for qRT-PCR in our laboratory was used. The RNA standard was quantified with a Qubit^®^ 2.0 fluorometer (Thermo Fisher Scientific, Waltham, MA, USA) for the preparation of serial dilutions. We found that RT-ddPCR results from serial dilutions were systematically lower than those expected from the Qubit^®^ 2.0 fluorometer. If the expected concentrations from the Qubit^®^ 2.0 fluorometer were used to determine LLOQ, the quantification range would be too narrow. Thus, we decided to use anticipated concentrations as proposed by Persson et al. [26] to determine LLOD95 and LLOQ. LLOQ of RT-ddPCR (217.02 copies/reaction) was higher than that of qRT-PCR (10 copies/reaction). We found that RT-ddPCR tended to report concentrations lower than those expected from dilution factors as previously reported [30]. The inefficiency of the reverse transcription might be one of the possible reasons [31]. We found that reporting quantification as absolute copies resulted in large variances which prohibited LLOQ determination. In addition, viral load is often reported and statistically analyzed as log10-copies due to its wide range and skewed distribution. Therefore, DENV RNA quantification results were analyzed as log10-copies in all of our subsequent analyses. LLOD95 and LLOQ values proposed here could be more precisely determined by conducting tests with additional dilutions. In this study, we defined LLOQ with relatively strict criteria [27]. However, more relaxed criteria for LLOQ, such as defining %CV cut-off at 25% [32], 30% [33], or 35% [25], could be applied to this protocol to widen the quantification range.

The ability of RT-ddPCR to detect DENV was superior to that of qRT-PCR. Since LLOD95 of RT-ddPCR was greater than that of qRT-PCR (700 copies/mL or 2.895 log10-copies/mL), the better sensitivity of RT-ddPCR was unexpected. One possible explanation is that the partitioning of reactions in RT-ddPCR increases the template concentration in each droplet resulting in higher amplification efficiency and tolerance to potential inhibitors in the sample [34]. This effect is expected to be more pronounced at very low RNA template concentrations similar to our findings from Bland–Altman plots (Figure 4B and Appendix A). In addition, RT-ddPCR consistently reported more DENV RNA copies in samples at low concentrations compared to qRT-PCR (Figure 4A). The superior detection and quantification performance of RT-ddPCR over qRT-PCR with environmental and clinical samples containing low target concentrations was also reported in previous studies on rotavirus [13], pepper mild mottle virus [35], pepino mosaic virus [36], and severe acute respiratory syndrome coronavirus 2 [37]. The results from three out of eight DENV samples detected by RT-ddPCR, but not qRT-PCR, were above LLOQ making the quantification results reportable. RT-ddPCR failed to detect RNA in only one DENV sample detected by qRT-PCR. RT-ddPCR actually reported 3.594 log10-copies/mL of DENV RNA in this sample which was slightly lower than the LLOD95 cut-off. Therefore, the detection result could not be reported with confidence. The specificity of RT-ddPCR tested with non-dengue samples was equal to that of qRT-PCR. RT-ddPCR reported noise in five out of 40 OFI samples (12.5%) ranging from 0.757 to 2.331 log10-copies/mL. Again, further optimization of LLOD95 with additional dilutions would improve the assay’s sensitivity. A limit of blank (LOB) may be derived with additional OFI samples to prevent noise reporting. We also performed serotype-specific analyses for sensitivity. The sensitivities of RT-ddPCR for detecting DENV1 and DENV3 were higher than those of qRT-PCR. However, the difference was not deemed statistically significant due to small serotype-specific sample sizes. We calculated the sample size for this study to confidently measure specificity and serotype-specific sensitivity of RT-ddPCR. However, we did not expect that the sensitivity of RT-ddPCR would be significantly higher than qRT-PCR. Thus, the sample size was not considered for comparing serotype-specific sensitivities between the two methods.

One major advantage of RT-ddPCR is that there is no requirement for a standard curve, which enables absolute quantification, unlike relative quantification by qRT-PCR [13]. Therefore, quantification results should be highly reproducible, and interlaboratory variations should be small. This was confirmed by DENV quantification results by RT-ddPCR from three laboratories (SI, BIOTEC and AFRIMS). We found that some specimens had quantification results exceeding ULOQ and were not included in interlaboratory variation analyses. Repeated measurements of these specimens were not conducted due to limited quantities of samples. For the future application of this RT-ddPCR protocol with clinical specimens, an additional 10-fold dilution for each clinical specimen should be prepared since several studies have reported dengue patients with very high viral loads (>10^6^ copies/mL), especially during the early days of illness [38,39].

RT-ddPCR might help answer important questions in the field of DENV research. A few studies reported incomplete genome or defective particles of DENV in clinical samples [40,41]. Multiplex RT-ddPCR might help characterize the defective genomes by simultaneously quantitating multiple regions of the same DENV RNA molecule. Currently, we are optimizing the multiplex RT-ddPCR protocol to accurately quantify defective genomes in clinical specimens. Although several studies linked DENV levels in plasma to disease severity [39,42,43,44,45], some studies reported conflicting results [46,47,48,49]. This contradiction may be caused by interlaboratory variations of qRT-PCR so standardizing the quantification with RT-ddPCR might validate the correlation between DENV levels and disease severity. We analyzed the correlation of DENV levels and disease severity from the cross-sectional samples described in Section 3.4 and found a significant difference of DENV levels between DF and DHF (multivariate regression *p* < 0.001, Appendix A). However, the kinetics of DENV levels from longitudinal sample collections should be analyzed to verify the correlation between viral loads and disease severity as reported by Avirutnan et al. [44]. RT-ddPCR can also help accelerate multi-center drug or vaccine trials by eliminating the need for transferring specimens to a single central laboratory. Finally, we believe that an advance in RT-ddPCR technology will make it become more competitive to qRT-PCR resulting in a wider application in the field.

## Figures and Tables

**Figure 1 diagnostics-11-00639-f001:**
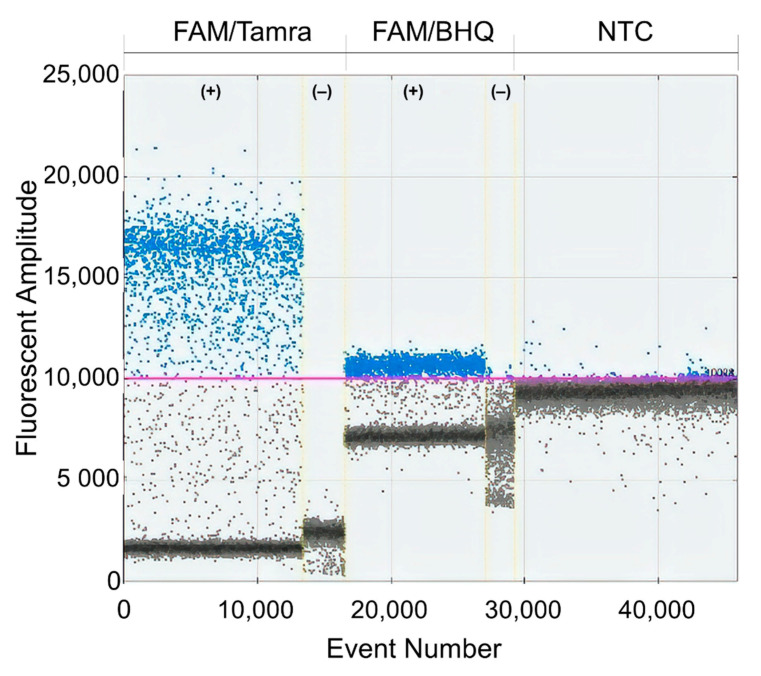
Comparison of FAM/Tamra and FAM/BHQ-labeled probes. 1D-plot results of FAM/Tamra and FAM/BHQ-labeled probes: X-axis represents the total number of droplets. Y-axis represents detected fluorescent amplitude with positive sample (+) by using in vitro RNA standard as a template. Negative sample (−) refers to the negative control (NTC) or blank where water was used as a template.

**Figure 2 diagnostics-11-00639-f002:**
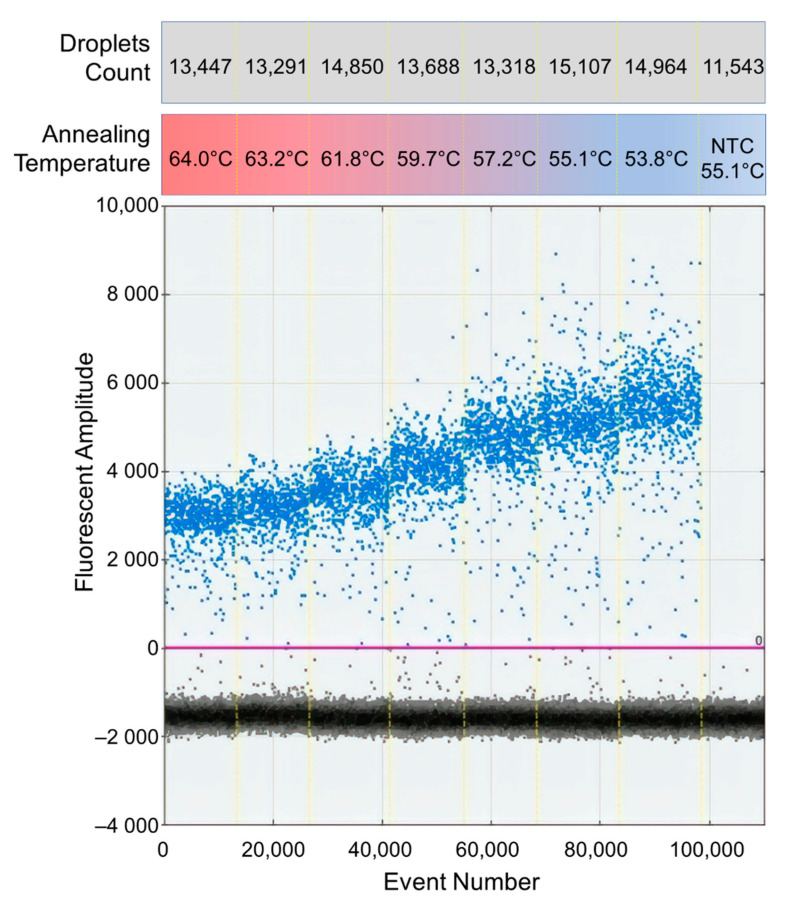
Temperature gradient PCR result of FAM/Tamra-labeled probes. The annealing temperature was varied from 53.8 °C to 64.0 °C. X-axis represents the total number of droplet events. Y-axis represents the detected fluorescent amplitude. NTC represents negative control. Total numbers of droplets count per reaction are shown in the gray box.

**Figure 3 diagnostics-11-00639-f003:**
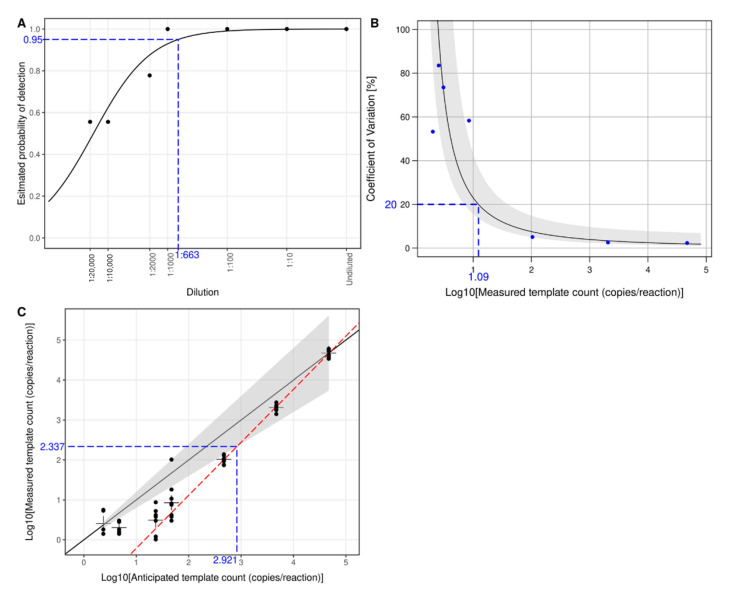
LLOD95 and LLOQ of RT-ddPCR for detection and quantification of DENV RNA. (**A**) LLOD95 was estimated by fitting a logistic regression model with detection results. Each dot indicates an actual detection rate at each dilution. The dashed horizontal line indicates 95% probability, and the dashed vertical line indicates an estimated dilution of 1:663 as LLOD95. LLOQ was determined by estimating %CV and comparing anticipated template counts to measured counts. (**B**) A precision profile shows a curve of estimated %CV and their corresponding confidence intervals (gray area). Each dot shows an actual %CV from measurements at the corresponding template count. The dashed horizontal line indicates a cut-off of %CV at ≤20%, and the dashed vertical line indicates a corresponding estimated template count of 1.090 log10-copies/reaction. (**C**) A comparison between anticipated and measured template counts shows measurement errors of RT-ddPCR. The solid diagonal line shows a perfect agreement, and the surrounding gray area indicates an acceptable error range of ±20% of the anticipated count. Each dot shows a measured template count at a corresponding anticipated count, and the “+” signs indicate average measured template counts. A linear regression analysis (red dashed line) using measured counts above LLOD95 is plotted against estimated LLOQ. The dashed horizontal line indicates an estimated LLOQ of 2.337 log10-copies/reaction, and the dashed vertical line indicates a corresponding anticipated template count of 2.921 log10-copies/reaction.

**Figure 4 diagnostics-11-00639-f004:**
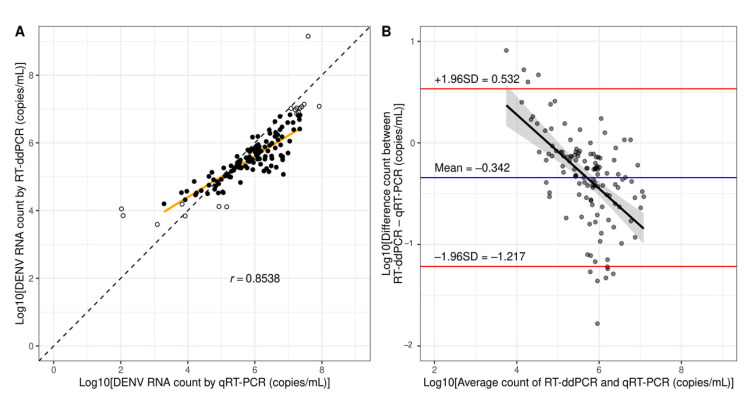
Comparison of quantification results between RT-ddPCR and qRT-PCR. (**A**) A regression line (orange) shows the correlation between dengue RNA from clinical samples measured by qRT-PCR and ddPCR. The diagonal line represents a hypothetical trend where results by the two methods are identical. The solid dots represent measured counts within the limit of quantification. The open dots represent measured counts below LLOQ or above ULOQ which were not included in the analysis. (**B**) Bland–Altman plot shows the limit of agreement between qRT-PCR and RT-ddPCR (red lines). The horizontal blue line shows the mean difference between the two methods (bias). A linear regression line and a confidence area show an association between measured counts and bias.

**Figure 5 diagnostics-11-00639-f005:**
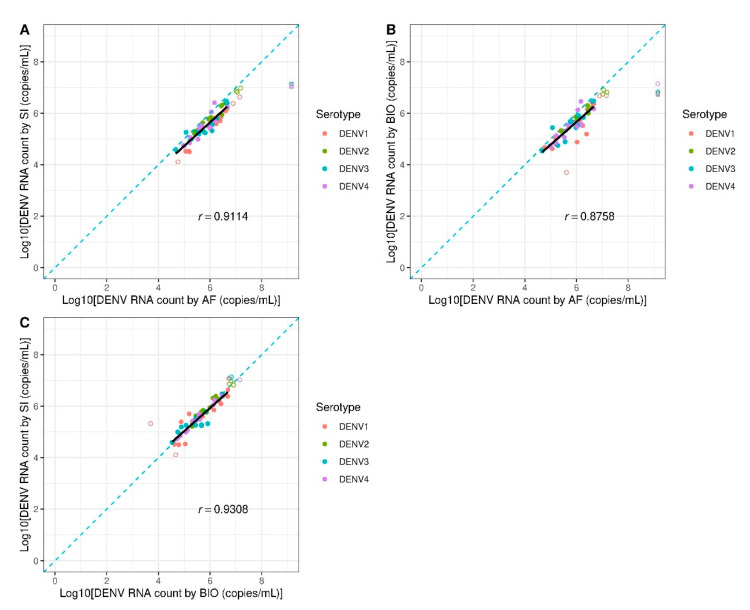
Linear regression analyses for interlaboratory variations. Regression lines (black lines) show the correlation of dengue RNA from clinical samples measured by RT-ddPCR between two laboratories: (**A**) SI versus AFRIMS (AF), (**B**) BIOTEC (BIO) versus AFRIMS (AF), and (**C**) SI versus BIOTEC (BIO). The solid dots represent measured counts within the limit of quantification. The open dots represent measured counts below LLOQ or above ULOQ which were not included in the analysis. The diagonal line (blue dashed line) represents a hypothetical trend where results from the two laboratories are identical.

**Figure 6 diagnostics-11-00639-f006:**
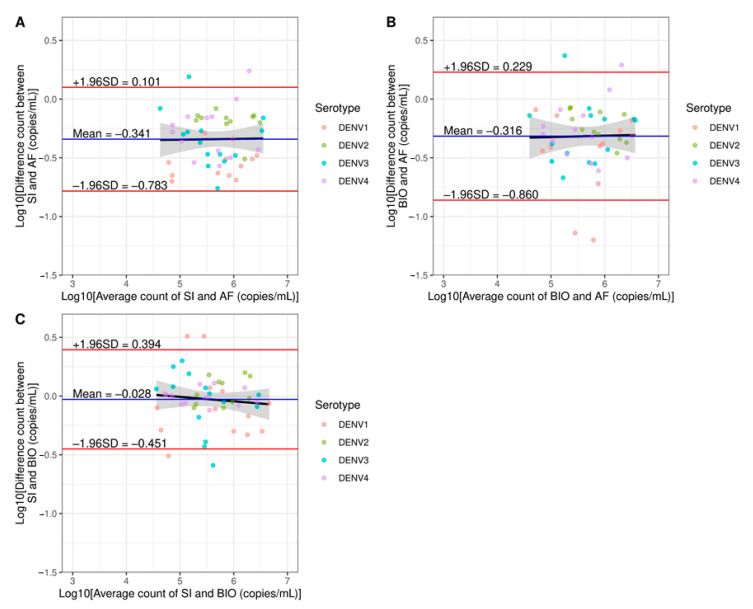
Bland–Altman plots for interlaboratory variations. Bland–Altman plots of RT-ddPCR results between two laboratories: (**A**) SI versus AFRIMS (AF), (**B**) BIOTEC (BIO) versus AFRIMS (AF), and (**C**) SI versus BIOTEC (BIO) with LOQ filtering. The horizontal black dashed lines represent agreement lines. The horizontal blue lines show mean differences between the two methods (bias). The horizontal red lines show agreement limits.

**Table 1 diagnostics-11-00639-t001:** DENV detection results of qRT-ddPCT and qRT-PCR with clinical samples.

		Positive qRT-PCR	Negative qRT-PCR
DENV Samples	Positive RT-ddPCR	140	8
(*n* = 156)	Negative RT-ddPCR	1	7
OFI Samples	Positive RT-ddPCR	0	0
(*n* = 40)	Negative RT-ddPCR	0	40

## Data Availability

All R scripts used for statistical analyses and result visualizations are publicly accessible at https://github.com/dummai/RT_ddPCR. The dataset used for calculating LLOD95 and LLOQ is shown in Appendix A. The datasets generated from clinical specimens during the current study are available from the corresponding authors on reasonable request.

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
