# Peer review of "Application of One-Step Reverse Transcription Droplet Digital PCR for Dengue Virus Detection and Quantification in Clinical Specimens"

_diagnostics, 2021, doi:10.3390/diagnostics11040639_

Round 1

Reviewer 1 Report

Materials and Methods. 2.3 - explain what Y stands for in the probe sequence. What type of fluorophore and quencher were used for the probe? 2.4- make sure that the units of concentration and/or volume are clear. 2.5 - should it be 10^5 copies/uL instead of 105 copies/uL. The same is true for the dilutions indicated in 2.6. Results. The LLOD calculations are a bit confusing. The authors report that the LLOD corresponded to 1:663 dilution, which would correspond to 10^5/663=150 .8 copies/reaction. Where 10^1.851 came from? Discussion. The authors report no false-positives when other flaviviruses were used, but do not present the data. It is recommended to include the data into the manuscript. It is also recommended to discuss the reported data (e.g. higher sensitivity of RT-ddPCR in comparison with that of qRT-PCR) in a context of the data available in the literature when the two methods were compared (e.g for DENV2 RNA or for other viral/bacterial etc RNA).

Reviewer 2 Report

I found this manuscript to be a very interesting read and one that is certainly worthy of publication in Diagnostics. This study developed and optimized a protocol for applying one-step RT-droplet digital PCR (RT-ddPCR) for DENV detection and quantification. They described RT-ddPCR as a better method than RT-qPCR in Dengue virus RNA detection in clinical specimens.

Few comments are listed below:

  1. Because the correlation of DENV levels in plasma to disease severity is unconcluded. But it would be interesting to know that whether the different level of DENV RNA is corrected to the cases with different disease severity in this study.
  2. Please provide the institutional review board (IRB) approval protocol number.
  3. Please correct the typos throughout the text.
